# Exploring differentially expressed genes between anagen and telogen secondary hair follicle stem cells from the Cashmere goat (*Capra hircus*) by RNA-Seq

Nimantana He[1,2], Rui Su[1,3,4,5], Zhiying Wang[1,5], Yanjun Zhang[1,5], Jinquan Li[1,3,4,5]*

**1** College of Animal Science, Inner Mongolia Agricultural University, Hohhot, Inner Mongolia, China, **2** Agriculture Research Center, Chifeng University, Chifeng, Inner Mongolia, China, **3** Key Laboratory of Animal Genetics, Breeding and Reproduction, Hohhot, Inner Mongolia Autonomous Region, China, **4** Key Laboratory of Mutton Sheep Genetics and Breeding, Ministry of Agriculture, Hohhot, China, **5** Engineering Research Center for Goat Genetics and Breeding, Hohhot, Inner Mongolia Autonomous Region, China

* lijinquan_nd@126.com

**Data Availability Statement:** The sequencing reads of each sequencing libraries have been deposited under NCBI with BioProject ID PRJNA588350 (BioSample: SAMN13241935,

## Abstract

Hair follicle stem cells (HFSCs) have been shown to be essential in the development and regeneration of hair follicles (HFs). The Inner Mongolia Cashmere goat (*Capra hircus*) has two types of HFs, primary and secondary, with cashmere being produced from the secondary hair follicle. To identify the genes associated with cashmere growth, transcriptome profiling of anagen and telogen secondary HFSCs was performed by RNA-Seq. The RNA-Seq analysis generated over 58 million clean reads from each group, with 2717 differentially expressed genes (DEGs) detected between anagen and telogen, including 1500 upregulated and 1217 downregulated DEGs. A large number of DEGs were predominantly associated with cell part, cellular process, binding, biological regulation and organelle. In addition, the PI3K-Akt, MAPK, Ras and Rap1 signaling pathways may be involved in the growth of HFSCs cultured *in vitro*. The RNA-Seq results showed that the well-defined HFSC signature genes and cell cycle-associated genes showed no significant differences between anagen and telogen HFSCs, indicating a relatively quiescent cellular state of the HFSCs cultured *in vitro*. These results are useful for future studies of complex molecular mechanisms of hair follicle cycling in cashmere goats.

## Introduction

Hair follicles, as skin appendages, play crucial role in skin homeostasis, and are also important in thermal regulation, social communication and sensory processes. In the 1980s, the location and characteristics of hair follicle stem cells were first identified using isotope labeling in mice [1], and subsequent studies have suggested that HFSCs play crucial roles in HF morphogenesis and cycling [2,3]. The results of intensive research in stem cell biology suggest that HFSCs are an excellent cell lineage to study stem cell plasticity [4], since most adult stem cells lose their

RNA-seq of cashmere goat anagen HFSC, SRA: SRR10417580).

**Funding:** Nimantana He, 2017M613263XB, project funded by China Postdoctoral Science Foundation; Nimantana He, 2019BS03012, project funded by Natural Science Foundation of Inner Mongolia; Jinquan Li, DC1900004124, project funded by Inner Mongolia Agricultural University. The funders had no role in study design, data collection and analysis, decision to publish or preparation of the manuscript.

**Competing interests:** The authors have declared that no competing interests exist.

self-renewal capacity and irreversibly differentiate, whereas hair follicles undergo continuous cycling throughout the lifetime. The morphogenesis and regeneration of hair follicle is a well-organized, spatiotemporal mesenchymal-epithelial interaction that requires complex signal transmission between the dermis and epidermis [5,6]. Hair follicle morphogenesis can be divided into three phases: induction, organogenesis and cytodifferentiation. During induction, Wnt mediated signal transduction first arises in mesenchymal cells directing the thickening of overlying epithelial cells to form a placode. The organogenesis stage consists of complex interplay of signals between the epidermis and the underlying dermis; epithelial placode cells direct the underlying dermal cells to proliferate and form a dermal condensate, which in turn signals the epithelial cells to proliferate and grow downwards into the dermis. In cytodifferentiation, the dermal condensate is enveloped with follicular epithelial cells thus forming distinct dermal papilla, which instruct the ectoderm to shape the entire HF through the action of morphogens and growth factors [7,8]. Based on the morphological characteristics, the hair cycle is divided into three major stages, including anagen (growth phase), catagen (regression phase) and telogen (resting phase) [9,10]. During anagen, bulge stem cell progeny populates the lower outer root sheath (ORS) towards the bulb matrix, constituting a specialized population of highly proliferative transit-amplifying (TA) cells and produce the hair shaft and the inner root sheath (IRS). Hair growth stops when matrix cells exhaust their proliferative capacity and transited into catagen phase. The apoptosis of matrix cells leads to the destruction of the lower two-thirds of the follicle, including the IRS, ORS, and hair bulb, whereas the bulge region remains intact. Catagen is followed by telogen phase. HFs remain quiescent, with the dermal papilla remaining in close proximity to the bulge throughout the telogen phase. When hair growth activating signals are generated, HFSCs are activated and initiate a new hair cycle [11–13].

In case of skin injury, bulge HFSCs become active and rapidly adopt differentiation programs that differ from normal homeostasis and involve the upward migration of HFSCs to repair damaged epidermis *in vivo*. These cells have been speculated to be tightly regulated, as they cease activity once the wound is healed [14, 15].

The Inner Mongolia Cashmere goat has two types of HFs that have a similar structure but differ in fineness and produce two distinct fibers, guard hairs produced by primary HFs and soft down produced by secondary HFs (Fig 1). Cashmere is produced from secondary HFs. The secondary hair follicles which exhibit a notable photoperiod-based cycle that changes throughout the year, are suitable for hair cycle research. Histological studies on secondary hair follicles from Inner Mongolia Cashmere goats have shown that anagen lasts from May to December, catagen from December to January, and telogen until the end of April [16]. To further investigate the molecular mechanism of hair follicle cycling, it is important to perform gene expression profiling of secondary hair follicle stem cells.

Transcriptome research is the basis of gene function and structure research. Through the high-throughput sequencing, almost all transcriptome information of a specific tissue or organ in a certain state can be obtained quickly and comprehensively. RNA-Seq is a commonly used high-throughput sequencing method used to detect differences in gene expression between two groups, which has been widely used in the fields of basic research and clinical diagnosis.

In our previous work, we isolated and identified primary HFSCs from Inner Mongolia Cashmere goat [17, 18]. In the present study, we used RNA-Seq to perform a genome-wide expression analysis of Inner Mongolia Cashmere goat HFSCs in the anagen and telogen phase to identify differentially expressed genes (DEGs). The results of this study will be useful for molecular and cellular analyses of hair follicle cycling.

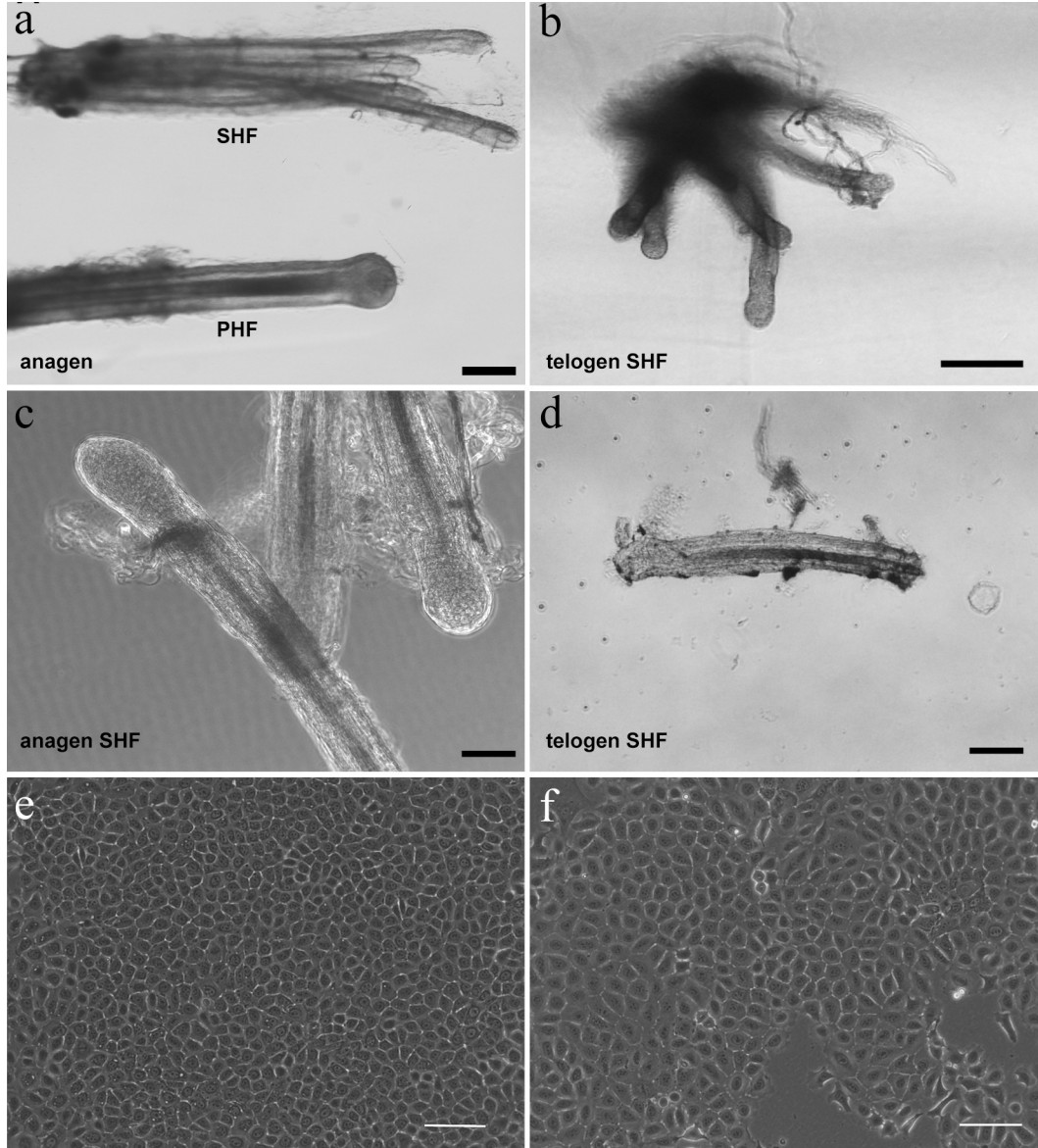

**Fig 1. Culture of Inner Mongolia Cashmere goat secondary HFSCs.** (a): Inner Mongolia cashmere goat primary (PHF) and secondary (SHF) hair follicles. (b, d): telogen phase SHF. (c): SHF from anagen phase. (e): in vitro cultured secondary HFSCs of anagen phase. (f): in vitro cultured secondary HFSCs of telogen phase. Scale bars 100 μm.

## Materials and methods

### Ethics statement

In this study, adult Arbas Cashmere goat skin samples used for cell isolation were collected in accordance with the International Guiding Principles for Biomedical Research Involving Animals and was approved by the Animal Ethics Committee of the Inner Mongolia Academy of Agriculture and Animal Husbandry Sciences (approval number IMAAAHS#1215000046002373XP) that is responsible for Animal Care and Use in the Inner Mongolia Autonomous Region of China. In our study, no specific permissions were required for these activities and the animals did not involve endangered or protected species. The catagen phase of the Inner Mongolia Cashmere goat

hair follicle is much shorter than anagen and telogen phase, and it is hard to isolate the cells. We focused on the differentially expressed genes in growth and quiescence state, so we collect anagen and telogen phase HFs for cell culture.

## Cell isolation

Anagen- and telogen-phase HFSCs (ana-SHFSCs and tel-SHFSCs) were isolated as previously described [17]. Briefly, HFs were separated from skin samples after digested by dispase and adhered to collagen IV-treated dishes for culturing. After tissue adherence, the medium (DMEM/F12 supplemented with 5% FBS, 1% penicillin–streptomycin mixture, 10 ng/mL EGF, 10 ng/mL insulin and 0.4 μg/mL hydrocortisone) was changed every 3 days, and cell growth was observed microscopically. Telogen HFs were treated as the same with the anagen HFs. Immunocytochemistry staining of widely used HFSC markers like Krt15, Krt19 and Sox9 was performed to identify the cells [19,20]. In brief, cells were fixed in 4% paraformaldehyde for 10 min, and permeabilized with 0.1% Triton X-100 in PBS for 10 min. Blocked with 10% goat serum for 1 h at room temperature. Primary antibodies against Krt15, Krt19 and Sox9 (1:300) were added separately to the cells, and incubated at 37 for 1 h. Control was incubated with 10% goat serum instead of primary antibodies. After washing, cells were incubated with secondary antibody (1:500) in the dark for 45 min. The nuclei were counterstained with DAPI (1:1000) for 5 min in the dark. The cells were visualized using a confocal microscope (A1, Nikon, Tokyo, Japan). Antibodies were purchased from Abcam (Cambridge, MA, USA).

## RNA extraction and quality analysis

Total RNA was extracted from approximately $1.0 \times 10^7$ ana-SHFSCs and $1.0 \times 10^7$ tel-SHFSCs from the second passage using TRIzol reagent (Invitrogen, USA) following the manufacturer's instructions. The RNA concentration, purity and integrity were determined using a micro-spectrophotometer NanoDrop 2000 (IMPLEN, CA, USA), a Qubit® 3.0 Flurometer (Life Technologies, CA, USA), and an Agilent RNA 6000 Nano Kit (Agilent Technologies, CA, USA), respectively.

## Library preparation and sequencing

After quality determination, mRNA was enriched with oligo (dT) magnetic beads and then broken into 350-bp fragments in fragmentation buffer. Using the mRNA fragments as templates, first strand cDNA was synthesized using random hexamer primers. Subsequently, dNTPs, RNase H, DNA polymerase I and buffer were added to synthesize second cDNA. Double-stranded cDNA was purified using a QiaQuick PCR kit, and eluted with EB buffer for end repair and single nucleotide adenine (A) addition. Finally, adaptors were ligated to the fragments. The target fragments were purified and PCR amplified to construct the library, which was sequenced on an Illumina platform.

## Mapping to the reference genome

High quality clean reads were obtained from the raw reads by removing low quality and adapter-contaminated reads. The clean reads were used for subsequent analysis and were aligned to the NCBI goat reference genome for *Capra hircus* ARS1. Bowtie2 v2.2.3 was used for building the genome index, and Clean Data was then aligned to the reference genome using HISAT2 v2.1.0. HISAT2 is the successor to TopHat2, which uses a modified BWT algorithm to convert reference to index for faster speed and fewer resources. The distribution in the gene region refers to the number and proportion of unique sequences aligned to the three

functional elements of the genes (exons, introns and intergenic regions) according to the annotation file.

## Expression annotation

For gene expression analysis, the number of uniquely matched reads was calculated and normalized to FPKM (fragments per kilobase per million mapped fragments). The level of expression for each gene in the two groups was compared to identify differences in expression using DEGseq as described by Wang et al., 2010 [21]. DEGseq is proposed based on MA-plot and widely used for differential gene expression analysis. The P-value could be assigned to each gene and adjusted by the Benjamini and Hochberg's approach for controlling the false discovery rate. Genes with q≤0.05 and |log2_ratio|≥1 are identified as differentially expressed genes (DEGs). Cluster analysis of gene expression patterns was performed by Hierarchical Cluster, and showed as heat map. In this study, we compared anagen to telogen, that is anagen data as sample, telogen data as control.

GO (Gene Ontology) enrichment analysis could exhibits the biological functions of the DEGs. The GO (http://geneontology.org/) enrichment of DEGs was implemented by the hypergeometric test, in which p-value is calculated and adjusted as q-value, and data background is genes in the whole genome. GO terms with q<0.05 were considered to be significantly enriched. GO functional enrichment analysis was performed by Blast2GO. Pathway enrichment analysis was carried out using KEGG (Kyoto Encyclopedia of Genes and Genomes) database. KEGG (http://www.kegg.jp/), a database resource containing a collection of manually drawn pathway maps representing our knowledge on the molecular interaction and reaction networks. The KEGG enrichment of DEGs was implemented by the hypergeometric test, in which p-value was adjusted by multiple comparisons as q-value. KEGG terms with q<0.05 were considered to be significantly enriched.

## Validation of RNA-Seq data

Total RNA was extracted from ana-SHFSs and tel-SHFSCs using a TRIzol Plus RNA Purification Kit (Invitrogen) following the manufacturer's protocols. The concentration of RNA was measured using a UV spectrophotometer (NanoDrop 2000, Thermo Scientific, Hudson, NH, USA), and reverse transcription to cDNA. Six differentially expressed genes were selected randomly for validation of RNA-Seq data. qRT-PCR was performed using an ABI 7300 real-time PCR system (Applied Biosystems, Foster City, CA, USA) with a SYBR Premix Ex Taq II kit (Takara, Dalian, China). The primers used for qRT-PCR are listed in Table 1, and β-actin was used as a reference gene. The qRT-PCR thermocycling parameters were as follows: 95˚C for 30 s followed by 40 cycles of 95˚C for 30 s and 60˚C for 30 s. The specificity of the SYBR green PCR signal was confirmed by melting curve analysis. The relative level of mRNA expression for each gene from triplicate experiments was calculated using the $2^{-\Delta\Delta Ct}$ method.

## Results

### Cell culture

Cell morphology of the anagen and telogen secondary HFSCs were alike. The cells showed typical morphology of hair follicle stem cells, including a cobblestone and nest appearance, high refractive index, good adhesion ability, small cell size, centralized, round and large nuclei with more than two nucleoli (Fig 1).

Immunocytochemistry staining showed that Krt15, Krt19 and Sox9 were positively expressed in the ana-SHFSCs and tel-SHFSCs (Fig 2).

**Table 1. Primer sequences for qRT-PCR.**

| gene | NCBI accession | sequence | |
|---|---|---|---|
| ACTB | NM_001314342.1 | forward | TCCTGCGTCTGGACCTGG |
| | | Reverse | CCTTGATGTCACGGACGATTC |
| LHX2 | XM_005881527.2 | forward | GATGCGGAGCACCTGGAC |
| | | reverse | TCGGGGTTGTGGTTAATGG |
| SOX9 | XM_012109896.2 | forward | ACAAGTTCCCCGTCTGCATC |
| | | reverse | GTGCGGCTTGTTCTTGCTC |
| LGR5 | XM_005679712.3 | forward | GTAGCTGGCTGATCCGAATTG |
| | | reverse | CCCATGAGCATGTTGACCG |
| RUNX2 | XM_012664647.1 | forward | TGGACGAGGCAAGAGTTTCAC |
| | | reverse | CTTCTGGGTTCCCGAGGTC |
| BMP4 | NM_007554.3 | forward | TCCCAAGCATCACCCACAG |
| | | reverse | GCCACAATCCAATCATTCCAG |
| KLF4 | XM_006079408.2 | forward | GTCCCACCGCTCCATTACC |
| | | reverse | ACCGCCTTCCCCTCTTTG |
| SLC6A6 | XM_012114225.2 | forward | CAAGGGGTGGACATTGCTG |
| | | reverse | CACTTCCACAAACTGGCTATCC |
| KRTAP3-1 | NM_001285774.1 | forward | CCTGCCACCACCATCTGC |
| | | reverse | CACAGAAAGTGGGCTGGAGG |

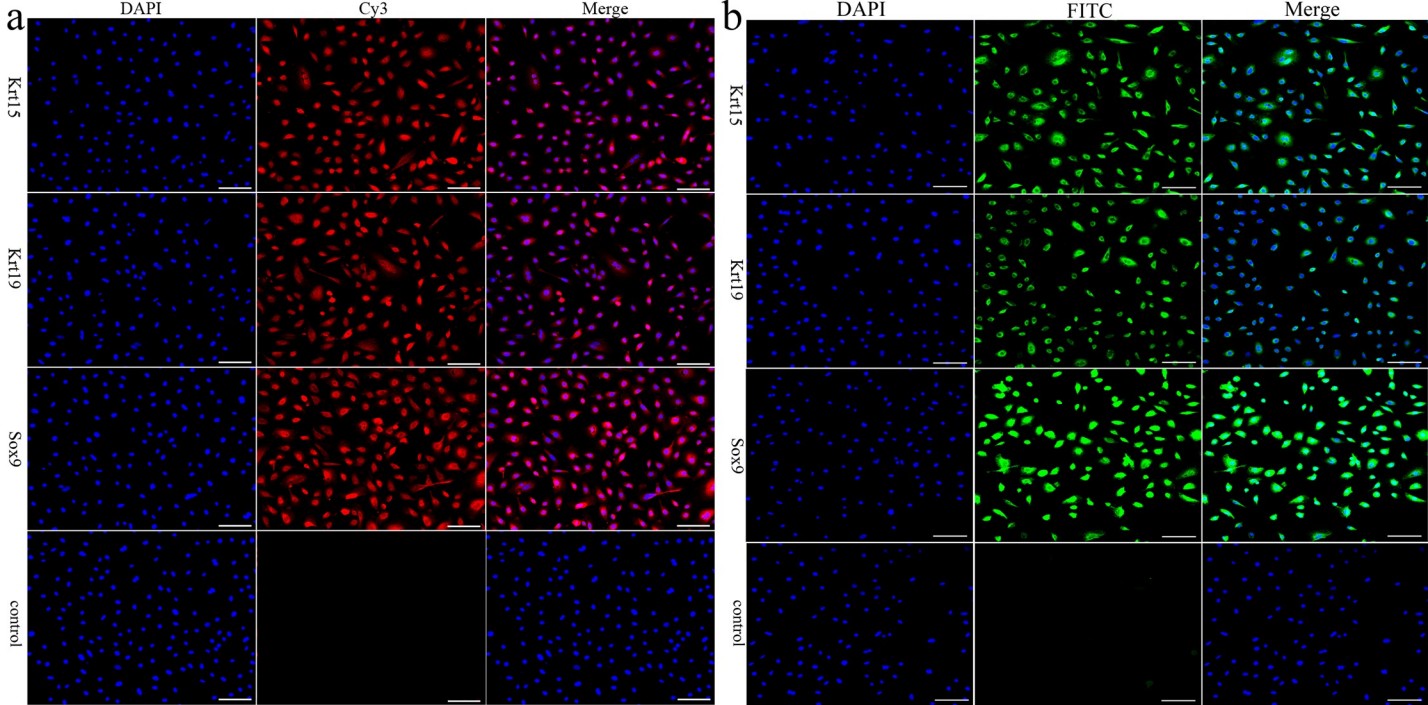

**Fig 2. Identification of the in vitro cultured HFSCs.** (a): Immunocytochemistry staining of Krt15, Krt19 and Sox9 in the anagen secondary HFSCs. (b): Immunocytochemistry staining of Krt15, Krt19 and Sox9 in the telogen phase secondary HFSCs. Scale bars 100 μm. Control was HFSCs incubated with 10% goat serum instead of primary antibody; red, Cy3–conjugated goat anti-rabbit IgG; green, Fluorescein (FITC)–conjugated goat anti-rabbit IgG; blue, DAPI staining.

## Alignment to the reference genome

To elucidate the gene expression patterns of ana-SHFSCs and tel-SHFSCs, we constructed cDNA libraries for the two independently cultured cell lines and performed deep sequencing using an Illumina HiSeq 2000 platform, resulting in 64,858,310 and 60,223,076 raw reads, respectively. After filtering out the adaptors and low-quality sequences, 63,393,498 and 58,734,944 clean reads were generated from the ana-SHFSCs and tel-SHFSCs representing 97.74 and 97.53% of the raw reads, respectively. Of these reads, 61,558,263 and 56,797,731 could be mapped to the goat reference genome (NCBI, Capra_hircus_ARS1, ftp://ftp.ncbi. nlm.nih.gov/genomes/all/GCF/001/704/415/GCF_001704415.1_ARS1/GCF_001704415.1_ARS1_genomic.fna.gz, and annotation file, ftp://ftp.ncbi.nlm.nih.gov/genomes/all/GCF/001/704/415/GCF_001704415.1_ARS1/GCF_001704415.1_ARS1_genomic.gff.gz), with mapping rates of 97.11 and 96.7%, respectively (Table 2).

## Gene expression profiles between anagen and telogen secondary HFSCs

The expression levels of individual genes from two groups were compared to assess differences in expression using DEGSeq. In this study, we compared anagen RNAseq data to telogen RNAseq data. It is like anagen data as sample, telogen data as reference/control.

From both the ana-SHFSCs and tel-SHFSCs, 16,849 transcripts were identified as being expressed genes from the FPKM values, of which 2717 genes were significantly differentially expressed between the two groups, including 1500 upregulated and 1217 downregulated genes (Fig 3, S1 Table), respectively.

Interestingly, among the well-known HFSC signature genes reported to be important for hair follicle stem cell biology, including *SOX9*, *LHX2*, *NFATC1*, *FGF18*, *RUNX1* and *VDR* [22], *LHX2* and *NFATC1* were observed to be differentially expressed in the RNA-Seq data (S2 Table). Most of the cell cycle genes, such as *CCNA2*, *CCNB2*, *CCNB1*, *CDKN1A* and *MCM5* were all expressed but did not exhibit differences in expression (S2 Table).

Previous studies have shown that KRT and KRTAP are major structural proteins of the hair fiber and sheath, and their content is important for fleece quality [23]. We identified a set of keratins (*KRT*) and keratin-associated protein (*KRTAP*) genes that were also annotated in cashmere goat (Capra hircus) hair follicles [24] (Table 3). Most of the differentially expressed *KRT* and *KRTAP* were down regulated in the telogen phase HFSCs.

## Functional classification

To better understand the HF cycle, DEGs were categorized into three gene ontology categories: cellular components, biological processes, and molecular functions. DEGs between the ana-

**Table 2. Summary of mapped read and mapping rates based on the RNA-Seq data.**

|  | ana-SHFSCs | tel-SHFSCs |
|---|---|---|
| Raw reads | 64858310 | 60,223,076 |
| Clean reads | 63,393,498 | 58,734,944 |
| Clean reads rate (%) | 97.74% | 97.53%. |
| Mapped Reads | 61,558,263 | 56,797,731 |
| Mapping Rate (%) | 97.11% | 96.7% |
| UnMapped Reads | 1,835,235 | 1,937,213 |
| MultiMap Reads | 2,775,221 | 3,284,301 |
| MultiMap Rate (%) | 4.38% | 5.59% |

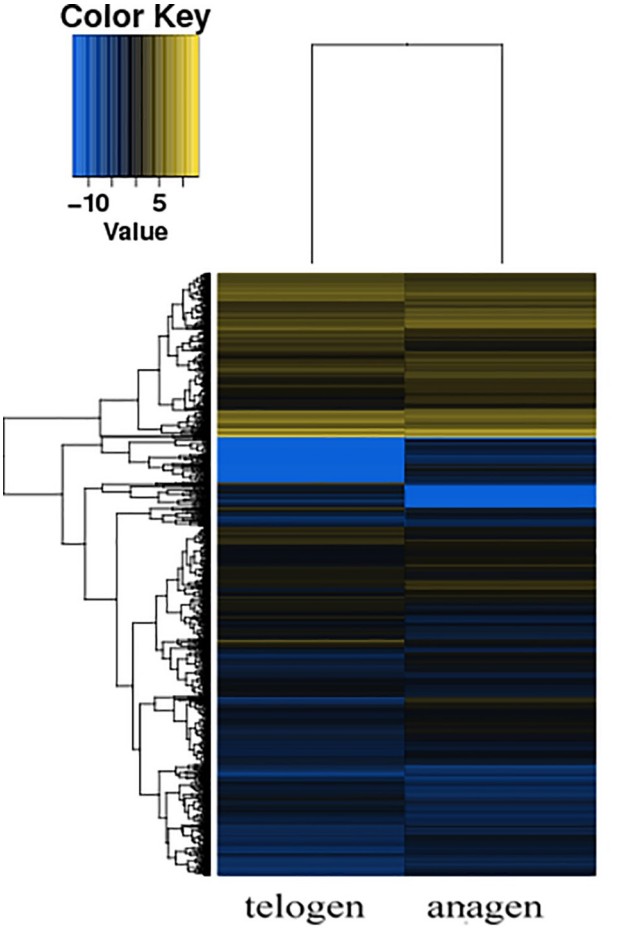

**Fig 3. Hierarchical cluster analysis of gene expression based on log ratio FPKM data.** Yellow indicates the genes with greater expression, and blue indicates the genes with lower expression. There were clusters with relatively minor differences between anagen and telogen secondary HFSCs.

SHFSCs and tel-SHFSCs were categorized into 55 functional groups based on sequence homology. In the three main GO classification categories, 26, 16, and 13 functional groups were identified, respectively (Fig 4). The top five functional categories for the upregulated and downregulated DEGs included cell part, cellular process, binding, biological regulation and organelle. Among these groups, the terms cell part, cellular process and binding were dominant in each of the three categories, respectively. We also noticed a high percentage of genes in membrane part, metabolic process, organelle part, catalytic activity, and developmental process. The GO analysis results showed that the functions of the identified DEGs were involved in various biological processes, such as cellular process, biological regulation, metabolic process, developmental process and response to stimulus, and were associated with HFs and skin development. For example, the DEGs *LHX2*, *LGR5* [25] and *FOXC1* [26], which were identified as being important in hair follicle development, were enriched in transcriptional activator, hair morphogenesis and hair cycle process categories, respectively.

The KEGG analysis results predicted that the DEGs were significantly enriched in pathways such as PI3K-Akt, MAPK, Ras and Rap1 (Table 4).

**Table 3. *KRT* and *KRTAP*, which were annotated in Cashmere goat (*Capra hircus*) hair follicles.**

| Gene | Fold Change(ana-tel) | Description |
|---|---|---|
| *KRT4* | 63.777962 | PREDICTED: keratin, type II cytoskeletal 4 [Capra hircus] |
| *KRT79* | 12.391147 | PREDICTED: keratin, type II cytoskeletal 79 isoform X2 [Ovis aries musimon] |
| *KRT7* | 6.381597 | PREDICTED: keratin, type II cytoskeletal 7 isoform X3 [Ovis aries musimon] |
| LOC102180595 | 4.1000119 | PREDICTED: keratin, type I cytoskeletal 18 isoform X2 [Ovis aries musimon] |
| LOC102179515 | 3.1967979 | PREDICTED: keratin, type I cytoskeletal 15 [Ovis aries musimon] |
| *KRT8* | 2.1871825 | PREDICTED: keratin, type II cytoskeletal 8 [Capra hircus] |
| LOC102180424 | 0.3971521 | PREDICTED: keratin, type I cytoskeletal 16 [Capra hircus] |
| *KRT1* | 0.3814496 | PREDICTED: keratin, type II cytoskeletal 1 isoform X2 [Ovis aries musimon] |
| *KRT78* | 0.3057293 | PREDICTED: keratin, type II cytoskeletal 78 isoform X1 [Capra hircus] |
| *KRTAP11-1* | 0.2014041 | keratin-associated protein 11–1 [Capra hircus] |
| *KRT23* | 0.1741835 | PREDICTED: keratin, type I cytoskeletal 23 [Capra hircus] |
| *KRT82* | 0.1518523 | PREDICTED: keratin, type II cuticular Hb2 isoform X1 [Capra hircus] |
| *KPRP* | 0.1447063 | PREDICTED: keratinocyte proline-rich protein-like [Capra hircus] |
| *KRT10* | 0.1368225 | PREDICTED: keratin, type I cytoskeletal 10 [Pantholops hodgsonii] |
| *KRTAP3-1* | 0.0759261 | keratin-associated protein 3–1 [Capra hircus] |
| LOC102182669 | 0.0364445 | PREDICTED: keratin, type II cytoskeletal 5-like [Ovis aries] |
| LOC102176685 | 0.0206982 | PREDICTED: keratin, type II cytoskeletal 75 isoform X2 [Ovis aries musimon] |
| LOC108635997 | 0.0200734 | PREDICTED: keratin, type II cytoskeletal 6A [Capra hircus] |

## qRT-PCR validation of the RNA-Seq data

To validate the results from the transcriptomic analysis, eight differentially expressed genes were randomly selected and assayed by qRT-PCR. When anagen compared to telogen (anagen as sample, telogen as control), the qRT-PCR results verified that these genes were differentially expressed in HFSCs at the anagen and telogen phases, consistent with the RNA-Seq results (Fig 5). Thus, the RNA-Seq results provided reliable data for the mRNA differential expression analysis.

## Discussion

Because of differences between the breed, heredity and living environment of domestic animals, our molecular and morphological understanding of hair follicle biology relies a great deal on mouse hair follicle research. In recent years, studies on Cashmere goat hair follicles has increased [27–29], providing valuable information for further research on the molecular mechanisms associated with these cells.

In this study, we performed gene expression profiling between anagen and telogen hair follicle stem cells of the Inner Mongolia Cashmere goat. With respect to the HFSC signature genes, we observed that *LHX2*, a master regulator of HFSC quiescence, was differentially expressed. *LHX2* has been shown to govern both cellular quiescence and differentiation [30]. We observed higher *LHX2* expression in anagen HFSCs in our data, indicating that *LHX2* is also associated with the activity of HFSCs, consistent with the results of a previous study [31–33]. *SOX9*, a master regulator of HFSCs, orchestrates the dynamics of cell fate decision, stem cell plasticity and the active-quiescent transition of HFSCs [34–36]. In our previous study, we

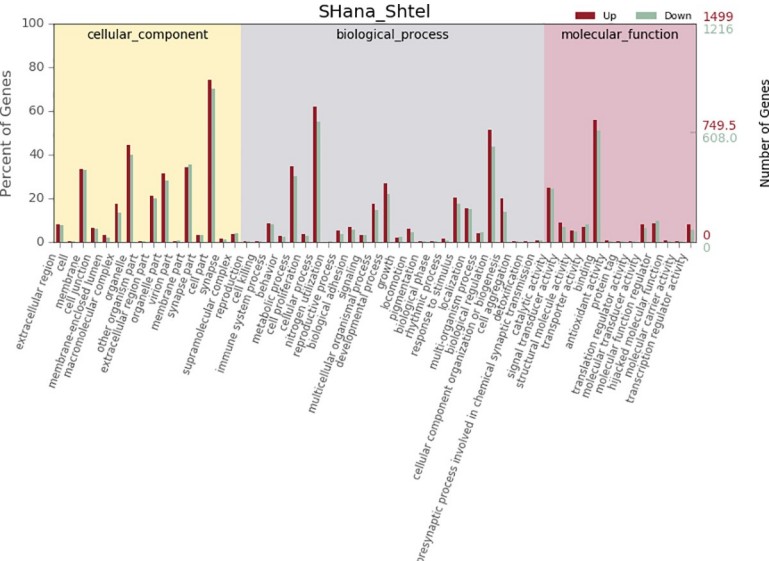

**Fig 4. GO classification of DEGs.** The results are summarized in three main categories: biological process, cellular component and molecular function. The X-axis indicates the second level term of gene ontology; The Y-axis shows the percentage of genes. red, up regulated genes; green, down regulated genes.

showed that *SOX9* is crucial in HFSC feature maintenance [37]. In this study, no significant difference in *SOX9* expression was observed between anagen and telogen HFSCs. Combining with mouse and human HFSC research [38,39], we hypothesize that during the distinct anagen and telogen phases of the Inner Mongolia Cashmere goat hair follicle, the HFSCs were under a

**Table 4. KEGG pathway analysis of DEGs.**

| Pathway term | FDR | Up Count | Down Count |
|---|---|---|---|
| MAPK signaling pathway | 1.806E-05 | 38 | 17 |
| Protein digestion and absorption | 1.806E-05 | 20 | 7 |
| PI3K-Akt signaling pathway | 2.128E-05 | 47 | 21 |
| Ras signaling pathway | 0.0001668 | 31 | 16 |
| Rap1 signaling pathway | 0.0002316 | 30 | 10 |
| Serotonergic synapse | 0.0008405 | 16 | 13 |
| Drug metabolism—cytochrome P450 | 0.0013631 | 8 | 13 |
| Thyroid hormone synthesis | 0.0015981 | 11 | 8 |
| ECM-receptor interaction | 0.0020869 | 15 | 4 |
| Relaxin signaling pathway | 0.0023643 | 20 | 7 |
| Focal adhesion | 0.0042909 | 25 | 9 |
| Aldosterone synthesis and secretion | 0.0048675 | 16 | 2 |
| Arachidonic acid metabolism | 0.0048675 | 9 | 10 |
| Melanogenesis | 0.0048675 | 15 | 6 |
| Regulation of lipolysis in adipocytes | 0.0048675 | 8 | 6 |
| Signaling pathways regulating pluripotency of stem cells | 0.0048675 | 16 | 10 |
| TNF signaling pathway | 0.0048675 | 16 | 6 |
| Glutathione metabolism | 0.0071094 | 4 | 11 |
| Calcium signaling pathway | 0.0080587 | 22 | 10 |
| Basal cell carcinoma | 0.0085833 | 11 | 4 |

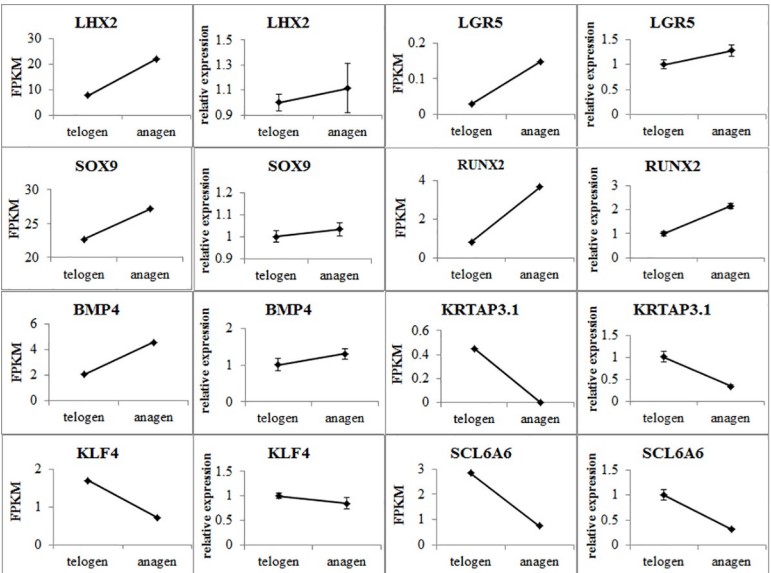

**Fig 5. qRT-PCR validation of the RNA-Seq data.** The qRT-PCR data are shown as the means ± standard error (SE) of three replicates. FPKM values from the RNA-Seq analysis are shown as the means and SE of three replicates. The left side shows the qRT-PCR data, while the right side shows the FPKM values from the RNA-Seq results.

relatively cellular quiescent state, but when activated at the initiation of the hair cycle or in response to injury, HFSCs begin to differentiate, indicating that *SOX9* plays a crucial role in HFSC maintenance. In agreement with this result, the cell cycle-related genes in our data showed no significant difference, indicating that the cells were in a relatively quiescent cellular state.

Hair follicle development and homeostasis are maintained by various signaling pathways. In our data, DEGs were significantly enriched in pathways that included the MAPK, PI3K-Akt, Ras, and Rap1 signaling pathways, which regulate the pluripotency of stem cells and cytokine-cytokine receptor interaction as well as cell adhesion molecules (CAMs). The PI3K pathway has been suggested to potentially regulate the transition of HFSCs from proliferation to quiescence [40]. Signaling pathways such as Wnt, TGF-β, Hedgehog and Notch, which are important in hair follicle development [41], were not significantly enriched in this study, possibly because the HFSCs were cultured *in vitro*, and the significantly enriched genes and pathways primarily contribute to the cellular process, maintaining HFSC features rather than developmental processes. Hair growth is a highly complex biological process that is affected by the environment, heredity, nutrition, cytokines and hormones. HFSCs reside in a distinct microenvironment, balancing stem cell maintenance with neighboring cells, and because extracellular matrix and signals are derived from these compartments, it is hard to mimic the *in vivo* microenvironment under *in vitro* culture conditions. Therefore, an *in vitro* culture system that can simulate the *in vivo* microenvironment is needed. In addition, further investigation into the molecular mechanism of the hair cycle is also needed.

From this study, we showed that the Inner Mongolia Cashmere goat HFSCs maintained a relatively cellular quiescent *in vitro*, in agreement with the quiescent nature of HFSCs [42, 43]. Maintaining cellular quiescence may allow cells to retain stemness or a high enough cell population for self-renewal and regeneration throughout the lifetime. Key challenges in the future are to identify the molecular mechanisms that control the hair follicle cycle, the dynamics between quiescence and stemness, and HFSC homeostasis. The results obtained using anagen

and telogen phase secondary HFSCs in our study will provide new data related to gene expression profiles in hair follicle cycling in cashmere goats.

## Supporting information

**S1 Table. DEGs in ana-SHFSCs and tel-SHFSCs.**
(XLSX)

**S2 Table. List of reported HFSC signature genes and reported cell cycle genes identified from the RNA-Seq data.**
(XLSX)

## Acknowledgments

We are grateful to the Inner Mongolia white Cashmere goat farm for the kind help and cooperation during the animal experiments.

## Author Contributions

**Formal analysis:** Rui Su.

**Investigation:** Nimantana He, Zhiying Wang.

**Methodology:** Zhiying Wang, Yanjun Zhang.

**Supervision:** Jinquan Li.

**Validation:** Yanjun Zhang.

**Writing – original draft:** Nimantana He.

**Writing – review & editing:** Rui Su.

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
