## [Decision Letter · Decision Letter 0]

9 Oct 2019

PONE-D-19-19805

Exploring differentially expressed genes between anagen and telogen secondary hair follicle stem cells of Cashmere goat (capra hircus) by RNA-seq

PLOS ONE

Dear Dr. He,

Thank you for submitting your manuscript to PLOS ONE. After careful consideration, we feel that it has merit but does not fully meet PLOS ONE’s publication criteria as it currently stands. Therefore, we invite you to submit a revised version of the manuscript that addresses the points raised during the review process.

You only randomly selected 6 genes and used the qPCR for RNA seq data validation. You should use qPCR to validate all important genes that you reference in your manuscript discussion section. Additinally,  instead of listing all the differentially expressed genes between the anagen HFSC and telogen HFSC (Table 2-4), present this  differentially expressed gene info as a heat map graph with genes in the same category clustered together. Please, also address the reviewers comments included below.

We would appreciate receiving your revised manuscript by November 15, 2019. To enhance the reproducibility of your results, we recommend that if applicable you deposit your laboratory protocols in protocols.io, where a protocol can be assigned its own identifier (DOI) such that it can be cited independently in the future. For instructions see: http://journals.plos.org/plosone/s/submission-guidelines#loc-laboratory-protocols

We look forward to receiving your revised manuscript.

Kind regards,

Irina Polejaeva, PhD

Academic Editor

PLOS ONE

Journal Requirements:

1. Thank you for including your ethics statement: Adult Arbas Cashmere goat was obtained from the Inner Mongolia YIWEI white Cashmere Goat Farm. Skin samples using in cell isolation were collected in accordance with the Animal Ethics Committee of the Inner Mongolia Academy of Agriculture and Animal Husbandry Sciences that is responsible for Animal Care and Use in the Inner Mongolia Autonomous Region of China.

Please amend your current ethics statement to confirm that your named ethics committee specifically approved this study.

For additional information about PLOS ONE submissions requirements for animal ethics, please refer to http://journals.plos.org/plosone/s/submission-guidelines#loc-animal-research  

2. We note that you are reporting an analysis of a microarray, next-generation sequencing, or deep sequencing data set. PLOS requires that authors comply with field-specific standards for preparation, recording, and deposition of data in repositories appropriate to their field. Please upload these data to a stable, public repository (such as ArrayExpress, Gene Expression Omnibus (GEO), DNA Data Bank of Japan (DDBJ), NCBI GenBank, NCBI Sequence Read Archive, or EMBL Nucleotide Sequence Database (ENA)). In your revised cover letter, please provide the relevant accession numbers that may be used to access these data. For a full list of recommended repositories, see http://journals.plos.org/plosone/s/data-availability#loc-omics or http://journals.plos.org/plosone/s/data-availability#loc-sequencing.

Reviewers' comments:

Reviewer's Responses to Questions

**Comments to the Author**

1. Is the manuscript technically sound, and do the data support the conclusions?

Reviewer #1: Partly

Reviewer #2: Partly

2. Has the statistical analysis been performed appropriately and rigorously? 

Reviewer #1: N/A

Reviewer #2: N/A

3. Have the authors made all data underlying the findings in their manuscript fully available?

Reviewer #1: Yes

Reviewer #2: Yes

4. Is the manuscript presented in an intelligible fashion and written in standard English?

Reviewer #1: Yes

Reviewer #2: Yes

5. Review Comments to the Author

Reviewer #1: In this article, the authors compared transcriptome profiles between anagen and telogen secondary HFSCs using RNA-Seq technique, and found 2717 differentially expressed genes (DEGs). The RNA-Seq results were verified by RT-PCR, in which 6 DEGs selected randomly were amplified and analyzed. The data in this article are obviously useful for researchers working in this field. The article is organized and written in standard English. The data have not been published elsewhere.

Abstract

1. Line 4-6: Cashmere only grows in the second hair follicles (HFs). Both anagen and telogen HFSCs were isolated from the second HFs (the authors did not state how to isolate HFSCs at different stages); therefore, the results in this paper can not disclose the differentially expressed genes between primary and secondary HFs.

2. Next page, Line 3: ‘in agreement with the quiescent nature of HFSCs’ is confusing.

Introduction

1. There are 3 important stages in hair cycle, growth (anagen), regression (catagen), and relative quiescence (telogen). The authors analyzed gene expressing profiles in both anagen and telogen stages. Why not include data in the catagen stage?

2. Page2, Last sentence: ‘Cashmere is the primary commercial product of the Inner Mongolia Cashmere goat, the secondary hair follicles from which exhibit a notable photoperiod-based cycle that changes throughout the year, making them suitable for hair cycle research.’ I suggest this sentence should be improved.

3. Page3: ‘RNA-Seq is a commonly used high-throughput sequencing method used to detect differences in gene expression between two groups.’ There is only one sentence in this paragraph. I suggest the authors either combine this sentence with last paragraph or include more information concerning RNA-Seq technique.

Materials and Methods

1. 1st paragraph: The isolation and identification of ana-SHFSCs and tel-SHFSCs should be written in a single paragraph with detailed descriptions. Are there any markers that can be used to identify the ana-SHFSCs and tel-SHFSCs?

2. Check the sentence: ‘The clean reads were used in for subsequent analysis and were aligned to the NCBI goat reference genome …’. The ‘in’ should be deleted.

Results

1. Please provide the isolation and identification results in the main text, not in the Supplement. How did the authors differentiate ana-SHFSCs and tel-SHFSCs in their exp.., with immunohistochemistry staining?

2. The figure S1: Some pictures in S1 Fig. look like those published in Reference 16, such as, morphology of HFSCs, Krt15 and Krt19 staining. Also there are no pictures showing different stages of HFSCs.

3. Check the sentence: ‘resulting in 64,858,310 and 60,223,076 raw reads being obtained, respectively (Table 1).’. The ‘being obtained’ should be deleted.

4. The figure S2: A typo in ‘Percentage’.

Discussion

1. Please provide the reference for this sentence, ‘indicating that LHX2 is also associated with the activity of HFSCs, consistent with the results of a previous study’. Also briefly introduce the results in the paper published.

2. No data support the conclusion in this sentence, ‘but when activated at the initiation of the hair cycle or in response to injury, HFSCs begin to differentiate, indicating that SOX9 plays a crucial role in HFSC maintenance.’. Please provide the reference and briefly introduce the results in the paper published.

3. It the ‘PI3K-Akt’ the same as the ‘PI(3)K’?

Reviewer #2: In this manuscripts, the authors aimed to investigate the transcriptome differences anagen and telogen secondary hair follicle stem cells in goats. They found 2717 genes were differentially expressed in these two populations of hair follicle stem cells. I found this type of study intriguing since it could provide a lot of information that not only could contribute to the cashmere production but also to our knowledge of the regenerative characteristics of stem cell. However, this experiment only has the RNA sequencing data. It will be more convincing and useful if the authors could do some follow-up experiment related to the differentially expressed genes found using RNA seq.

1. The only data this manuscript have is the RNA seq data. To validate the RNA seq, the authors randomly picked 6 genes and did the qPCR. I suggest the authors use qPCR to validate all these important genes that you made your discussion on.

2. Table 2-4 listed all the differentially expressed genes between the anagen HFSC and telogen HFSC. I found this is a very inefficient and confusing way to present RNA seq data. I highly suggest the authors put this differentially expressed gene info into a heat map graph with genes in the same category clustered together. The results part included too much technique information that should be put into the methods section. For example, the summary of mapped read and mapping rates based on the RNA-Seq, seems really irrelevant to the study aim. In results, the authors to list the experiment outcome that contribute to test the hypothesis of the study.

3. Tabe3,4. I don’t think it’s necessary to mention they genes has different reads but has no significant difference.

6. PLOS authors have the option to publish the peer review history of their article (what does this mean?). If published, this will include your full peer review and any attached files.

Reviewer #1: No

Reviewer #2: No

---

## [Author Response · Author response to Decision Letter 0]

3 Dec 2019

Reviewer #1

Abstract

1. Line 4-6: Cashmere only grows in the second hair follicles (HFs). Both anagen and telogen HFSCs were isolated from the second HFs (the authors did not state how to isolate HFSCs at different stages); therefore, the results in this paper cannot disclose the differentially expressed genes between primary and secondary HFs.

In this paper, we were aimed to explore the differentially expressed genes between anagen and telogen phase of the secondary hair follicle, so there is no data or result about differentially expressed genes between primary and secondary HFs. We revised the statement. 

Isolation of the cells from anagen and telogen phase hair follicle was detailed in the Materials and Methods section.

2. Next page, Line 3: ‘in agreement with the quiescent nature of HFSCs’ is confusing.

This statement was deleted.

Introduction

1. There are 3 important stages in hair cycle, growth (anagen), regression (catagen), and relative quiescence (telogen). The authors analyzed gene expressing profiles in both anagen and telogen stages. Why not include data in the catagen stage?

The hair cycle of Inner Mongolia Cashmere goat can divided into anagen, catagen, and telogen three stages. We focused on the differentially expressed genes in growth and quiescence state. In addition, the catagen phase of the Inner Mongolia Cashmere goat hair follicle is much shorter than anagen and telogen phase, and it is hard to isolate the cells. So we chose anagen and telogen.

2. Page2, Last sentence: ‘Cashmere is the primary commercial product of the Inner Mongolia Cashmere goat, the secondary hair follicles from which exhibit a notable photoperiod-based cycle that changes throughout the year, making them suitable for hair cycle research.’ I suggest this sentence should be improved.

We improved the sentence as ‘The secondary hair follicles which exhibit a notable photoperiod-based cycle that changes throughout the year, are suitable for hair cycle research.’

3. Page3: ‘RNA-Seq is a commonly used high-throughput sequencing method used to detect differences in gene expression between two groups.’ There is only one sentence in this paragraph. I suggest the authors either combine this sentence with last paragraph or include more information concerning RNA-Seq technique.

We combined this sentence with last paragraph.

Materials and Methods

1. 1st paragraph: The isolation and identification of ana-SHFSCs and tel-SHFSCs should be written in a single paragraph with detailed descriptions. Are there any markers that can be used to identify the ana-SHFSCs and tel-SHFSCs?

The isolation of the ana-SHFSCs and tel-SHFSCs were detailed in the cell isolation. We are not sure if there are any markers that can be identify the ana-SHFSCs and tel-SHFSCs nowadays. In this paper, we use well-known HFSC markers to identify the cells.

2. Check the sentence: ‘The clean reads were used in for subsequent analysis and were aligned to the NCBI goat reference genome …’. The ‘in’ should be deleted.

Yes, the ‘in’ has been deleted.

Results

1. Please provide the isolation and identification results in the main text, not in the Supplement. How did the authors differentiate ana-SHFSCs and tel-SHFSCs in their exp.., with immunohistochemistry staining?

Thank you. Isolation and identification results were put in the main text. Cell morphology of ana-SHFSCs and tel-SHFSCs were much alike under microscope observation. Isolation, culture and RNA extraction of ana-SHFSCs and tel-SHFSCs were carried out separately, avoiding from mixed throughout the experiment.

2. The figure S1: Some pictures in S1 Fig. look like those published in Reference 16, such as, morphology of HFSCs, Krt15 and Krt19 staining. Also there are no pictures showing different stages of HFSCs.

I am the first author of the reference 16. We isolated HFSCs from primary and secondary (anagen and telogen) HFs, and carried out a series of experiments. The morphology of HFSCs from the primary and secondary (anagen and telogen) were with high similarity, and markers used to identify the HFSCs were the same, so we chose different secondary antibody (FITC and Cy3) in reference 16 and this paper.

3. Check the sentence: ‘resulting in 64,858,310 and 60,223,076 raw reads being obtained, respectively (Table 1)’. The ‘being obtained’ should be deleted.

It has been deleted.

4. The figure S2: A typo in ‘Percentage’.

Yes. Figure S2 is showed as percentage.

Discussion

1. Please provide the reference for this sentence, ‘indicating that LHX2 is also associated with the activity of HFSCs, consistent with the results of a previous study’. Also briefly introduce the results in the paper published.

Mardaryev AN, Meier N, Poterlowicz K, Sharov AA, Sharova TY, Ahmed MI, et al. Lhx2 differentially regulates Sox9, Tcf4 and Lgr5 in hair follicle stem cells to promote epidermal regeneration after injury. Development. 2011; 138:4843–4852. doi: 10.1242/dev.070284. PMID: 22028024

In this paper, they found that Lhx2+ cells reside in the stem cell-enriched epithelial compartments (bulge, secondary hair germ) and co-express selected stem cell markers (Sox9, Tcf4 and Lgr5). Cell proliferation in the bulge and the number of Sox9+ and Tcf4+ cells in the HFs closely adjacent to the wound in Lhx2+/– mice are decreased in comparison with wild-type controls. Chip-on-chip/ChIP-qPCR and reporter assay analyses identified Sox9, Tcf4 and Lgr5 as direct Lhx2 targets. These data strongly suggest that Lhx2 positively regulates Sox9 and Tcf4 in the bulge cells, and promotes wound re-epithelization. Thus, Lhx2 operates as an important regulator of epithelial stem cell activity in the skin response to injury. 

Törnqvist G, Sandberg A, Hägglund AC, Carlsson L. Cyclic expression of lhx2 regulates hair formation. PLoS Genet. 2010:8;6(4):e1000904. doi: 10.1371/journal.pgen.1000904. PMID: 20386748

In this paper, they found Lhx2 is primarily expressed by precursor cells outside of the bulge region where the HF stem cells are located. They hypothesis this developmental, stage- and cell-specific expression of Lhx2 regulates the generation and regeneration of hair. Moreover, transgenic expression of Lhx2 in postnatal HFs is sufficient to induce anagen. Thus, Lhx2 is an essential positive regulator of hair formation.

Combining the results in the papers mentioned above and our data, it suggests that Lhx2 is associated with the activity of the HFSCs. HFSC is a major cell population that participated in the morphogenesis and regeneration of the HFs.

2. No data support the conclusion in this sentence, ‘but when activated at the initiation of the hair cycle or in response to injury, HFSCs begin to differentiate, indicating that SOX9 plays a crucial role in HFSC maintenance’. Please provide the reference and briefly introduce the results in the paper published.

In our manuscript, the entire sentence is ‘We hypothesize that during the distinct anagen and telogen phases, the HFSCs were under a relatively cellular quiescent state, but when activated at the initiation of the hair cycle or in response to injury, HFSCs begin to differentiate, indicating that Sox9 plays a crucial role in HFSC maintenance.’ 

The following references may explain our conjectures, and we add these papers in the references.

Sox9 Is Essential for Outer Root Sheath Differentiation and the Formation of the Hair Stem Cell Compartment. Current Biology. 2005;15:1340–1351. DOI 10.1016/j.cub.2005.06.064)

Using tissue-specific inactivation of Sox9, the authors demonstrate that this gene serves a crucial role in hair differentiation and that skin deleted for Sox9 lacks external hair. Sox9 knock hair show severe proliferative defects and the stem cell niche never forms. They summarized that Sox9 directs differentiation of the ORS and is required for the formation of the hair stem cell compartment.

Jonathan A. Nowak, Lisa Polak, H. Amalia Pasolli, and Elaine Fuchs. Hair Follicle Stem Cells Are Specified and Function in Early Skin Morphogenesis. Cell Stem Cell.2008; 3, 33–43. DOI 10.1016/j.stem.2008.05.009. 

From this paper they found that the progeny of Sox9-expressing cells contribute to all skin epithelial lineages and Sox9 is required for SC specification.

Meelis Kadaja, Brice E. Keyes, Mingyan Lin, H. Amalia Pasolli, Maria Genander,Lisa Polak, Nicole Stokes, Deyou Zheng, and Elaine Fuchs. SOX9: a stem cell transcriptional regulator of secreted niche signaling factors. Genes Dev. 2014; 28: 328-341. doi:10.1101/gad.233247.113

By conditionally targeting Sox9 in adult HFSCs, they show that SOX9 is essential for maintaining them. The findings reveal roles for SOX9 in regulating adult HFSC maintenance and suppressing epidermal differentiation in the niche.

Rene C. Adam, Hanseul Yang, Shira Rockowitz, Samantha B. Larsen, Maria Nikolova,Daniel S. Oristian, Lisa Polak, Meelis Kadaja, Amma Asare, Deyou Zheng, and Elaine Fuchs. Pioneer factors govern super-enhancer dynamics in stem cell plasticity and lineage choice Nature. 2015 May 21; 521(7552): 366–370. doi:10.1038/nature14289.

In this paper, it has been identified SOX9 as a crucial chromatin rheostat of HFSC super-enhancers, and provide functional evidence that super-enhancers are dynamic, dense TF-binding platforms which are acutely sensitive to pioneer master regulators whose levels define not only spatial and temporal features of lineage-status, but also stemness, plasticity in transitional states and differentiation.

Hair follicles (HFs) undergo cyclical periods of growth, which are fueled by stem cells (SCs) at the base of the resting follicle. HFSC formation occurs during HF development and Sox9 is essential for the HFSCs specification, maintenance and hair differentiation.

3. It the ‘PI3K-Akt’ the same as the ‘PI(3)K’?

Yes, they are the same, and we unify the writing as PI3K.

Reviewer #2

1. The only data this manuscript have is the RNA seq data. To validate the RNA seq, the authors randomly picked 6 genes and did the qPCR. I suggest the authors use qPCR to validate all these important genes that you made your discussion on.

Thank you very much. We appreciate your suggestion. When we carried out the qPCR, we just randomly chose, and just focused on the signature genes, so picked LHX2, LGR5, SOX9 and RUNX2. qPCR validation of SOX9 and RUNX2 was now added to the manuscript. 

We carried out qPCR to validate the RNA-seq data, and the qPCR result of those genes was in consistent with the RNA-seq data, and it provided reliable data for the mRNA differential expression analysis. What`s more, lacking of nucleotide sequences of capra hircus genes in the NCBI, so it is hard to validate all those genes we mentioned in the discussion. 

2. Table 2-4 listed all the differentially expressed genes between the anagen HFSC and telogen HFSC. I found this is a very inefficient and confusing way to present RNA seq data. I highly suggest the authors put this differentially expressed gene info into a heat map graph with genes in the same category clustered together. The results part included too much technique information that should be put into the methods section. For example, the summary of mapped read and mapping rates based on the RNA-Seq, seems really irrelevant to the study aim. In results, the authors to list the experiment outcome that contribute to test the hypothesis of the study.

We reformat the tables and maps as your suggestion.

3. Tabe3,4. I don’t think it’s necessary to mention they genes has different reads but has no significant difference.

We put table3,4 as supplemental material.

---

## [Decision Letter · Decision Letter 1]

23 Jan 2020

PONE-D-19-19805R1

Exploring differentially expressed genes between anagen and telogen secondary hair follicle stem cells of Cashmere goat (capra hircus) by RNA-seq

PLOS ONE

Dear Dr. He,

Thank you for submitting your manuscript to PLOS ONE. After careful consideration, we feel that it has merit but does not fully meet PLOS ONE’s publication criteria as it currently stands. Therefore, we invite you to submit a revised version of the manuscript that addresses the points raised during the review process.

You addressed most of the comments from the previous reviewers. However, there are still recommendations that were not addressed. Please carefully read review 3 comments and address them by either making changes in the manuscript or providing answers in a response to the reviewers document.

We would appreciate receiving your revised manuscript by March 1, 2020. To enhance the reproducibility of your results, we recommend that if applicable you deposit your laboratory protocols in protocols.io, where a protocol can be assigned its own identifier (DOI) such that it can be cited independently in the future. For instructions see: http://journals.plos.org/plosone/s/submission-guidelines#loc-laboratory-protocols

We look forward to receiving your revised manuscript.

Kind regards,

Irina Polejaeva, PhD

Academic Editor

PLOS ONE

Reviewers' comments:

Reviewer's Responses to Questions

**Comments to the Author**

1. If the authors have adequately addressed your comments raised in a previous round of review and you feel that this manuscript is now acceptable for publication, you may indicate that here to bypass the “Comments to the Author” section, enter your conflict of interest statement in the “Confidential to Editor” section, and submit your "Accept" recommendation.

Reviewer #1: All comments have been addressed

Reviewer #3: (No Response)

2. Is the manuscript technically sound, and do the data support the conclusions?

Reviewer #1: Yes

Reviewer #3: Partly

3. Has the statistical analysis been performed appropriately and rigorously? 

Reviewer #1: N/A

Reviewer #3: No

4. Have the authors made all data underlying the findings in their manuscript fully available?

Reviewer #1: Yes

Reviewer #3: Yes

5. Is the manuscript presented in an intelligible fashion and written in standard English?

Reviewer #1: Yes

Reviewer #3: Yes

6. Review Comments to the Author

Reviewer #1: All my comments have been addressed.

The authors did a great job. The results in this study are important to elucidate the molecular mechanisms of hair follicle cycling in goats.

Reviewer #3: In the revision of manuscript “Exploring differentially expressed genes between anagen and telogen secondary hair follicle stem cells of Cashmere goat (capra hircus) by RNA-seq”, the authors answered previous reviewers’ questions, but there are still several important issues need to be clearly addressed:

1. What’s the differences between hair follicle stem cells at anagen and telogen stages? Even though authors included the immunostaining of two different stages, but still no differences had been found/addressed.

2. In the other publication, “Exploring Differentially Expressed Genes by RNA-Seq in Cashmere Goat (Capra hircus) Skin during Hair Follicle Development and Cycling (Geng et al. Plos one 2013, reference 25)”, same methods/strategies (RNA-Seq) were used as current study to investigate differentiation during three stages of hair follicle development. Have authors compared the similar/differences of gene expression at same stage between current and previous study? Also, as mentioned in the manuscript “Cashmere is the primary commercial product of the Inner Mongolia Cashmere goat” ,how would current study potentially help cashmere production in the future, prolong the anagen stage?

3. Methods and software used for statistical analysis in this study is not clear.

4. In Results, most of description is confused whether authors compared differentiated expression between two stages of HFSCs, or compared two stages HFSCs to other stem cells.

Specific comments:

1. Please add page number and line number in the manuscript, which is easier for reviewer to refer.

2. Please list the full name of all abbreviations, when first mentioned, e.g. GO, KEGG. Please check the whole manuscript.

3. Introduction, paragraph 2, “During anagen… the telogen stage”, could be shorten. Since this study is focused on HFSCs, so a little more information about how they have been activated and initiate hair follicle cycling should be included.

4. “Cashmere is the primary commercial product of the Inner Mongolia Cashmere goat”, it’s better refer Cashmere is produced from secondary HFs, for readers who don’t know.

5. I think authors should add one or two more sentences about RNA-seq technique.

6. M&M, Cell isolation, add reference after 1st sentence. Add references for HFSC markers- Krt15, Krt19 and Sox9, staining procedure, company of antibodies (or add references). Moreover, authors explained the reason of excluding catagen phase in this study, and I think this information should be included in M&M.

7. Statistical analysis should be separated part in M&M.

8. Results, please move all the legends to the end of text. Fig 2, what cells were used for control staining? add scale bar for Fig. 2.

9. Gene expression profiles in Results, “of which 2717 genes were significantly differentially expressed between the two groups”, I wonder all these 2717 genes were differentially expressed in which group of cells.

10. “Interestingly, among the well-known HFSC signature genes reported to be important for hair follicle stem cell biology, including SOX9, LHX2, NFATC1, FGF18, RUNX1 and VDR [19], LHX2 and NFATC1 were observed to be differentially expressed in the RNA-Seq data (Table S2). Most of the cell cycle genes, such as CCNA2, CCNB2, CCNB1, CDKN1A and MCM5 were all expressed but did not exhibit differences in expression (Table S2)”, again, were all these genes differentially expressed in anagen or telogen phase?

11. “The qRT-PCR results verified that these genes were differentially expressed in HFSCs at the anagen and telogen phases, consistent with the RNA-Seq results”, so these genes were differentially expressed in HFSCs at both stages? Then what was the reference sample?

12. Fig.4, different colors of column represent what?

13. “We hypothesize that during the distinct anagen and telogen phases of the Inner Mongolia Cashmere goat hair follicle, the HFSCs were under a relatively cellular quiescent state, but when activated at the initiation of the hair cycle or in response to injury, HFSCs begin to differentiate,”, any references to support? Or similar finding in other species?

7. PLOS authors have the option to publish the peer review history of their article (what does this mean?). If published, this will include your full peer review and any attached files.

Reviewer #1: No

Reviewer #3: No

---

## [Author Response · Author response to Decision Letter 1]

27 Feb 2020

Reply to the Reviewers

Thank you for your kindly and academic suggestions and advises. We made reply to reviewer#3 and specific comments below, and we revised the manuscript according to comments. 

Reviewer #3: 

In the revision of manuscript “Exploring differentially expressed genes between anagen and telogen secondary hair follicle stem cells of Cashmere goat (capra hircus) by RNA-seq”, the authors answered previous reviewers’ questions, but there are still several important issues need to be clearly addressed:

1. What’s the differences between hair follicle stem cells at anagen and telogen stages? Even though authors included the immunostaining of two different stages, but still no differences had been found/addressed.

We first obtained HFs at anagen and telogen phase, and cultured cells. In vitro cultured HFSCs from anagen and telogen positively expressed well defined HFSC markers, and with similar morphology. Then we carried out RNAseq. From the RNAseq data, we found that most of the genes associate with cell cycle and HFSC significant genes were with no differences. Functional categories for the DEGs (upregulated and downregulated) included cell part, cellular process, binding, biological regulation and organelle. From KEGG analysis, it is found that DEGs were enriched in PI3K-Akt, MAPK, Ras and Rap1 signaling pathways, and these pathways were mainly participated in the cellular process and biological regulation. Pathways like Wnt/β-catenin, TGF, and SHH, which are play roles in the development of hair follicle, were not significantly enriched. In our research, we explored DEGs in HFSC in its different status. We noticed a high percentage of genes in terms of membrane part, metabolic process, organelle part, catalytic activity, and developmental process.

2. In the other publication, “Exploring Differentially Expressed Genes by RNA-Seq in Cashmere Goat (Capra hircus) Skin during Hair Follicle Development and Cycling (Geng et al. Plos one 2013, reference 25)”, same methods/strategies (RNA-Seq) were used as current study to investigate differentiation during three stages of hair follicle development. Have authors compared the similar/differences of gene expression at same stage between current and previous study? Also, as mentioned in the manuscript “Cashmere is the primary commercial product of the Inner Mongolia Cashmere goat”, how would current study potentially help cashmere production in the future, prolong the anagen stage?

We have read reference 25(Geng et al. Plos one 2013). In our present paper, we did not compare the gene expression. 

Cashmere production is affected by many factors, such as breed, nutrition, and environment. We aimed to find differentially expressed genes in in vitro cultured bulge stem cells. If functional genes can be found out, we could try genetic approach to prolong the anagen phase.

In reference 25 the authors explored differentially expressed genes in skin samples according to the hair cycle, not in hair follicles. 

I found that many published articles on exploring genes which play important roles in cashmere goat hair follicle development or hair cycle all using skin sample. Skin is composed of epidermal, dermal and subcutaneous tissue. As an appendage of skin, hair follicle is defined as a mini organ with its stem cell niche in the bulge, and bulge stem cells are critical for the hair morphogenesis and hair cycle. So, I think using adult skin sample to explore differentially expressed genes in hair development is not specific. We know it is very hard to isolate hair follicle cells from domestic animal like cashmere goat and find wood sheep, and it is a limitation of in vitro study.

3. Methods and software used for statistical analysis in this study is not clear.

We added methods and software used for statistical analysis in the text. 

4. In Results, most of description is confused whether authors compared differentiated expression between two stages of HFSCs, or compared two stages HFSCs to other stem cells.

In this article, we compared differentiated expression between two stages of HFSCs.

Specific comments:

1. Please add page number and line number in the manuscript, which is easier for reviewer to refer.

Page number and line number were added in the manuscript.

2. Please list the full name of all abbreviations, when first mentioned, e.g. GO, KEGG. Please check the whole manuscript.

Full name of abbreviations was added when first mentioned.

3. Introduction, paragraph 2, “During anagen… the telogen stage”, could be shorten. Since this study is focused on HFSCs, so a little more information about how they have been activated and initiate hair follicle cycling should be included.

Information about hair follicle morphogenesis and hair cycle were added.

4. “Cashmere is the primary commercial product of the Inner Mongolia Cashmere goat”, it’s better refer Cashmere is produced from secondary HFs, for readers who don’t know.

It has been revised.

5. I think authors should add one or two more sentences about RNA-seq technique.

It has been added.

6. M&M, Cell isolation, add reference after 1st sentence. Add references for HFSC markers- Krt15, Krt19 and Sox9, staining procedure, company of antibodies (or add references). Moreover, authors explained the reason of excluding catagen phase in this study, and I think this information should be included in M&M.

References for markers and ICC procedure, and explanation of excluding catagen phase were added in the M&M.

7. Statistical analysis should be separated part in M&M.

Statistical analysis was included in each step of the M&M, and it is easy to understand how we analysis the data step by step. So, we tend to include statistical analysis in each step if it is not necessary. 

8. Results, please move all the legends to the end of text. Fig 2, what cells were used for control staining? add scale bar for Fig. 2.

All legends were moved to the end of text. Fig.2 was revised.

9. Gene expression profiles in Results, “of which 2717 genes were significantly differentially expressed between the two groups”, I wonder all these 2717 genes were differentially expressed in which group of cells.

10. “Interestingly, among the well-known HFSC signature genes reported to be important for hair follicle stem cell biology, including SOX9, LHX2, NFATC1, FGF18, RUNX1 and VDR [19], LHX2 and NFATC1 were observed to be differentially expressed in the RNA-Seq data (Table S2). Most of the cell cycle genes, such as CCNA2, CCNB2, CCNB1, CDKN1A and MCM5 were all expressed but did not exhibit differences in expression (Table S2)”, again, were all these genes differentially expressed in anagen or telogen phase?

11. “The qRT-PCR results verified that these genes were differentially expressed in HFSCs at the anagen and telogen phases, consistent with the RNA-Seq results”, so these genes were differentially expressed in HFSCs at both stages? Then what was the reference sample?

Explain 9-11 

In this paper, DEGs were shown as anagen compared to telogen. It is like anagen data as sample, telogen data as reference/control, and we revised in the manuscript.

12. Fig.4, different colors of column represent what?

In fig.4 up regulated genes in red, and down regulated genes in green. We added it in the fig.4 legend.

13. “We hypothesize that during the distinct anagen and telogen phases of the Inner Mongolia Cashmere goat hair follicle, the HFSCs were under a relatively cellular quiescent state, but when activated at the initiation of the hair cycle or in response to injury, HFSCs begin to differentiate,”, any references to support? Or similar finding in other species?

I read the article “Fuchs Elaine. Skin Stem Cells in Silence, Action, and Cancer. Stem Cell Reports. 2018; 10(5), 1432-1438”, and combined with our result get to the hypothesize that during the distinct anagen and telogen phases of the Inner Mongolia Cashmere goat hair follicle, the HFSCs were under a relatively cellular quiescent state. Some articles mention about the quiescent of HFSCs in the mouse and human are list below, and we add two of these to the Reference in the manuscript.

Fuchs Elaine. Skin Stem Cells in Silence, Action, and Cancer. Stem Cell Reports. 2018; 10(5), 1432-1438. doi:10.1016/j.stemcr.2018.04.008. PMID：29742389

Lay, K., Kume, T., and Fuchs, E. FOXC1 maintains the hair follicle stem cell niche and governs stem cell quiescence to preserve long-term tissue-regenerating potential. Proc. Natl. Acad. Sci. USA. 2016; 113(11), E1506-15. doi:10.1073/pnas.1601569113. PMID：26912458

Horsley Valerie, Aliprantis Antonios O., Polak Lisa, Glimcher Laurie H., Fuchs Elaine. NFATc1 balances quiescence and proliferation of skin stem cells. Cell. 2008; 132(2), 299-310. doi:10.1016/j.cell.2007.11.047. PMID：18243104

Lien Wen-Hui., Fuchs Elaine. Wnt some lose some: transcriptional governance of stem cells by Wnt/β-catenin signaling. Genes Dev. 2014; 28(14), 1517-1532. doi:10.1101/gad.244772.114. PMID：25030692

Adam Rene C., Yang Hanseul., Rockowitz Shira., Larsen Samantha B., Nikolova Maria., Oristian Daniel S., Polak Lisa., Kadaja Meelis., Asare Amma., Zheng Deyou., Fuchs Elaine. Pioneer factors govern super-enhancer dynamics in stem cell plasticity and lineage choice. Nature. 2015; 521(7552), 366-370. doi:10.1038/nature14289. PMID：25799994

Ge Yejing, Gomez Nicholas C., Adam Rene C., Nikolova Maria., Yang Hanseul., Verma Akanksha., Lu Catherine Pei-Ju., Polak Lisa., Yuan Shaopeng., Elemento Olivier., Fuchs Elaine. Stem Cell Lineage Infidelity Drives Wound Repair and Cancer. Cell. 2017; 169(4), 636-650.e14. doi:10.1016/j.cell.2017.03.042. PMID：28434617

---

## [Decision Letter · Decision Letter 2]

20 Mar 2020

PONE-D-19-19805R2

Exploring differentially expressed genes between anagen and telogen secondary hair follicle stem cells of Cashmere goat (capra hircus) by RNA-seq

PLOS ONE

Dear Dr. He,

Thank you for submitting your manuscript to PLOS ONE. After careful consideration, we feel that it has merit but does not fully meet PLOS ONE’s publication criteria as it currently stands. Therefore, we invite you to submit a revised version of the manuscript that addresses the points raised during the review process.

Please incorporate minor suggestions from the reviewer included below.

We would appreciate receiving your revised manuscript by March 24. To enhance the reproducibility of your results, we recommend that if applicable you deposit your laboratory protocols in protocols.io, where a protocol can be assigned its own identifier (DOI) such that it can be cited independently in the future. For instructions see: http://journals.plos.org/plosone/s/submission-guidelines#loc-laboratory-protocols

We look forward to receiving your revised manuscript.

Kind regards,

Irina Polejaeva, PhD

Academic Editor

PLOS ONE

Reviewers' comments:

Reviewer's Responses to Questions

**Comments to the Author**

1. If the authors have adequately addressed your comments raised in a previous round of review and you feel that this manuscript is now acceptable for publication, you may indicate that here to bypass the “Comments to the Author” section, enter your conflict of interest statement in the “Confidential to Editor” section, and submit your "Accept" recommendation.

Reviewer #3: All comments have been addressed

2. Is the manuscript technically sound, and do the data support the conclusions?

Reviewer #3: Yes

3. Has the statistical analysis been performed appropriately and rigorously? 

Reviewer #3: Yes

4. Have the authors made all data underlying the findings in their manuscript fully available?

Reviewer #3: Yes

5. Is the manuscript presented in an intelligible fashion and written in standard English?

Reviewer #3: Yes

6. Review Comments to the Author

Reviewer #3: The manuscript has been improved, and most of points have been addressed clearly. But I was a bit confused the two versions of manuscript in this submission. Some revision can’t be found in the first one, but only shown in the one with track on: for example, “telogen data as reference/control” could not been found in the clean version (this information should be included in the manuscript); Legend of figure were missing in the first one, too. Please check which is the final version for revision.

Besides that, I only have a few minor comments:

1. L123, provide country of Abcam products produced.

2. L167, move “Gene Ontology, http://geneontology.org/” to L166, after first GO mentioned in this paragraph.

3. L234-235, add reference.

4. All the names of genes should be italic. Please check the discussion part.

5. L292-293, any references to support SOX9 expression was correlated to differentiation of HFSCs, which caused by initiation of hair cycle or injury?

7. PLOS authors have the option to publish the peer review history of their article (what does this mean?). If published, this will include your full peer review and any attached files.

Reviewer #3: No

---

## [Author Response · Author response to Decision Letter 2]

21 Mar 2020

1.Some revision can’t be found in the first one, but only shown in the one with track on: for example, “telogen data as reference/control” could not been found in the clean version (this information should be included in the manuscript); Legend of figure were missing in the first one, too. Please check which is the final version for revision.

Sorry for this. I checked two version, and found that I was mistaken the track version as clean version. This time I got them right, and marked the revised section in the manuscript with track.

2.L123, provide country of Abcam products produced.

We added the provide country of antibodies from Abcam（Cambridge, MA，USA）.

3.L167, move “Gene Ontology, http://geneontology.org/” to L166, after first GO mentioned in this paragraph.

“Gene Ontology, http://geneontology.org/” in the line 167 was moved to the beginning of the paragraph after first GO mentioned.

4.L234-235, add reference.

There are many articles from 1990s on keratins and skin or hair follicle, and we just added one review in the reference list as “23. Lutz Langbein and Jürgen Schweizer. Keratins of the human hair follicle. Int Rev Cytol. 2005; 243:1-78. doi:10.1016/S0074-7696(05)43001-6. PMID: 15797458”.

5.All the names of genes should be italic. Please check the discussion part.

Yes, we check the entire manuscript, and revised all the gene names in italic.

6.L292-293, any references to support SOX9 expression was correlated to differentiation of HFSCs, which caused by initiation of hair cycle or injury?

References from 33-39 can support the hypothesis that expression of Sox9 is associated with the differentiation and activation of HFSCs during the hair follicle initiation stage or injury.

---

## [Editor Report · Decision Letter 3]

24 Mar 2020

Exploring differentially expressed genes between anagen and telogen secondary hair follicle stem cells of Cashmere goat (capra hircus) by RNA-seq

PONE-D-19-19805R3

Dear Dr. He,

We are pleased to inform you that your manuscript has been judged scientifically suitable for publication and will be formally accepted for publication once it complies with all outstanding technical requirements.

With kind regards,

Irina Polejaeva, PhD

Academic Editor

PLOS ONE
---

## [Editor Report · Acceptance letter]

6 Apr 2020

PONE-D-19-19805R3 

Exploring differentially expressed genes between anagen and telogen secondary hair follicle stem cells from the Cashmere goat (*Capra hircus*) by RNA-Seq 

Dear Dr. He:

I am pleased to inform you that your manuscript has been deemed suitable for publication in PLOS ONE. Congratulations! Your manuscript is now with our production department. 

With kind regards,

on behalf of

Dr Irina Polejaeva 

Academic Editor

PLOS ONE